# Genetically Encoded Fluorescent Sensor for Poly-ADP-Ribose

**DOI:** 10.3390/ijms21145004

**Published:** 2020-07-15

**Authors:** Ekaterina O. Serebrovskaya, Nadezda M. Podvalnaya, Varvara V. Dudenkova, Anna S. Efremova, Nadya G. Gurskaya, Dmitry A. Gorbachev, Artem V. Luzhin, Omar L. Kantidze, Elena V. Zagaynova, Stanislav I. Shram, Konstantin A. Lukyanov

**Affiliations:** 1Shemyakin-Ovchinnikov Institute of Bioorganic Chemistry, Miklukho-Maklaya 16/10, 117997 Moscow, Russia; katyaakts@gmail.com (E.O.S.); npodvalnaya@gmail.com (N.M.P.); ngurskaya@mail.ru (N.G.G.); igorbachev@icloud.com (D.A.G.); 2Institute of Molecular Genetics, Kurchatova Sq. 2, 123182 Moscow, Russia; anna.efremova.83@gmail.com (A.S.E.); shram@img.ras.ru (S.I.S.); 3Institute of Experimental Oncology and Biomedical Technologies, Privolzhsky Research Medical University, Minin and Pozharsky Sq. 10/1, 603005 Nizhny Novgorod, Russia; orannge@mail.ru (V.V.D.); ezagaynova@gmail.com (E.V.Z.); 4Institute of Translational Medicine, Pirogov Russian National Research Medical University, Ostrovityanova 1, 117997 Moscow, Russia; 5Institute of Gene Biology, Vavilova 34/5, 119334 Moscow, Russia; artyom.luzhin@gmail.com (A.V.L.); kantidze@gmail.com (O.L.K.); 6Lobachevsky State University of Nizhny Novgorod, Gagarin Ave. 23, 603950 Nizhny Novgorod, Russia

**Keywords:** DNA damage respose, WWE-domai, PAR, fluorescent protein, FRET, sensor

## Abstract

Poly-(ADP-ribosyl)-ation (PARylation) is a reversible post-translational modification of proteins and DNA that plays an important role in various cellular processes such as DNA damage response, replication, transcription, and cell death. Here we designed a fully genetically encoded fluorescent sensor for poly-(ADP-ribose) (PAR) based on Förster resonance energy transfer (FRET). The WWE domain, which recognizes iso-ADP-ribose internal PAR-specific structural unit, was used as a PAR-targeting module. The sensor consisted of cyan Turquoise2 and yellow Venus fluorescent proteins, each in fusion with the WWE domain of RNF146 E3 ubiquitin ligase protein. This bipartite sensor named sPARroW (sensor for PAR
relying on WWE) enabled monitoring of PAR accumulation and depletion in live mammalian cells in response to different stimuli, namely hydrogen peroxide treatment, UV irradiation and hyperthermia.

## 1. Introduction

Poly-(ADP-ribosyl)-ation (PARylation) is a reversible post-translational modification of proteins and DNA resulting in covalent attachment of poly-ADP-ribose (PAR) polymers to a variety of amino acid residues on target proteins [1,2], or to 5’- and 3’-terminal phosphate residues at double- and single-strand breaks of a DNA molecule [3,4,5]. As any other post-translational modification, PARylation alters biochemical and functional properties of target proteins and affects protein–protein or protein–nucleic acid interactions. Accumulating evidence indicates that PAR conveys cellular signals through direct binding to different protein motifs, contributing to various cellular processes such as the DNA damage response, replication, chromatin structure remodeling, transcription, telomere homeostasis, and cell death [6].

Despite high interest in PARylation that developed over several decades, tools for real time monitoring intracellular level of PAR in living cells are scarce. Antibody detection is widely used, but the need for fixation and permeabilization of the cells limits the temporal resolution of this method. It is particularly important given that PAR polymers are extremely short-lived molecules with half-lives of less than 1 minute [7]. A turn-on split luciferase sensor utilizing APLF PBZ domains for PAR-binding [8] allows detection of PAR in cell lysates, without single cell resolution available for microscopic techniques. A work by Buntz et al. [9] describes a method using fluorescent cell-permeable NAD^+^ analogue as a Förster resonance energy transfer (FRET) acceptor, and EGFP-labeled ARTD1 as a FRET donor to monitor PARylation of EGFP-ARTD1 with fluorescence lifetime imaging microscopy (FLIM). The drawback of this approach is the need for the exogenous ligand, which limits the applicability outside cell culture models. Very recently, PBZ (PAR-binding zinc finger) domains were used to construct genetically encoded fluorescent PAR sensor variants [10]. Translocation of PBZ tagged with a fluorescent protein was shown to highlight spots of PAR accumulation, whereas PBZ in combination with split GFP enabled a large-scale search for PARylated proteins.

In the present work we designed the first fully genetically encoded FRET-based sensor for PAR and demonstrated its applicability for dynamic detection of PAR accumulation and degradation in living cultured cells.

## 2. Results

We took advantage of a bipartite sensor, in which donor and acceptor fluorescent proteins (FP) are separate polypeptide chains, each containing also a PAR-binding domain. As PAR is a polymer, its interaction with PAR-binding domains would bring FPs to the distance where FRET occurs (Figure 1A). At the same time, fully separated donor and acceptor FPs would ensure nearly zero FRET in the resting state.

Among many known PAR-binding protein domains, WWE domains recognize iso-ADP-ribose (iso-ADPR), the smallest internal PAR structural unit containing the characteristic ribose–ribose glycosidic bond [11,12]. As a FRET pair, we chose popular cyan Turquoise2 [13] and yellow Venus [14] fluorescent proteins. Based on this FRET pair we designed sPARroW (sensor for PAR
relying on WWE)—a sensor consisting of Turquoise2-WWE and Venus-WWE fusion proteins with a flexible amino acid linker between fluorescent proteins and the WWE domain (Figure 1A and Appendix A).

To test its response to PAR-inducing stimuli, we first analyzed subcellular distribution of sPARroW and found that it accumulated in the nuclei of H_2_O_2_-treated cells, reaching peak nuc/cyto ratio (as calculated by acceptor signal intensity) 25 min after addition of 100 µM H_2_O_2_ (Figure 1B and Appendix A). This effect was abolished by the pretreatment with PARP inhibitor PJ34 at 10 µM concentration (Figure 1C and Appendix A).

Then we used ratiometric imaging of FRET efficiency that takes into account fluorescence intensity in three channels: donor channel, acceptor channel and FRET channel (donor excitation wavelength and acceptor detection range), see Methods for calculations. FRET efficiency increased in most cells after H_2_O_2_ treatment (Figure 1D and Appendix A). Notably, pre-treatment with PJ34 abolished this effect (Figure 1E and Appendix A), indicating that the FRET efficiency increase requires PARP-1/2 activity. Hence, ratiometric FRET efficiency measurement can be used as an indicator of PAR accumulation in the nucleus. Additionally, with a longer observation time we were able to detect the decline in both translocation and FRET efficiency after the initial rise (Figure 1B,D), highlighting the ability of sPARroW to follow both accumulation and depletion of PAR in living cells in real time. It was also possible to detect H_2_O_2_-dependent FRET efficiency change in U2OS cell line stably expressing sPARroW (Appendix A). Potential advantage of using stable expression is that cells with relatively low variability of donor and acceptor expression levels can be selected by fluorescence-activated cell sorting (FACS). However, we found that even with transient expression, donor/acceptor ratio is mostly uniform between individual cells and does not correlate with FRET efficiency (Appendix A). To verify the sPARroW response by an independent method, we used standard immunostaining with commercial polyclonal antibodies against PAR. Upon treatment with 100 µM H_2_O_2_, we detected PAR accumulation in individual cell nuclei that was abolished by pretreatment with PJ34 inhibitor (Appendix A). This behavior corresponded well to the sPARroW-based results.

Ratiometric FRET efficiency measurement was somewhat complicated by a change of local concentration of fluorophores caused by sensor translocation to the nucleus. Therefore, we designed a nuclear-localized variant of the sensor, sPARroW^NLS^, by fusing a nuclear localization signal to the C-terminus of fluorescent proteins. As expected, sPARroW^NLS^ did not change its subcellular distribution after H_2_O_2_ treatment. Additionally, we designed a control sPARroW-R163A^NLS^ variant with a mutation in WWE domain known to abolish PAR binding [15]. Then, we used FLIM to characterize all sensor variants, both capable and incapable of PAR binding. Two types of PAR-inducing treatment were applied: localized irradiation with 405 nm laser, or incubation with H_2_O_2_. Local laser irradiation is a standard way to induce localized DNA damage, which is often used as a means to evoke PAR synthesis [16]. We analyzed live cells using FLIM and detected a significant decrease of mean donor fluorescence lifetime from 4.0 to 3.7 ns, first within the irradiated area and then over the whole nucleus of the irradiated cell (Figure 2A). For the cells treated with H_2_O_2_, we detected a similar 0.4-ns decline in the mean fluorescence lifetime (measured in the nucleus) for sPARroW and sPARroW^NLS^, but not for sPARroW-R163A^NLS^ mutant (Figure 2B,C). We also detected a very small (less than 0.1 ns), but still statistically significant decline in fluorescence lifetime for donor-only expressing cells (the reason for this phenomenon is unclear).

To rule out the possibility of a FRET increase being due to the interaction of sPARroW with some unidentified PAR-binding protein, we measured FRET in solution after addition of synthetic PAR to the purified sPARroW. We measured fluorescence emission spectra of a mixture of Turquoise2-WWE (0.15 μM) and Venus-WWE (0.45 μM) before and after addition of PAR (0.1 μM final concentration) at a 434 nm excitation wavelength. Addition of PAR resulted in an expected change of shape and intensity of emission spectrum: about 25% decrease of the donor peak at 480 nm and relative increase of the shoulder of acceptor emission at 530 nm (Figure 3A). This was not the case for the control donor + acceptor mixture lacking PAR-binding WWE domains, where emission spectra remained constant (Figure 3B).

It is well established that in mammalian cells mild hyperthermia (heat shock) can induce single- and double-stranded DNA breaks, which activate the DNA damage response (DDR) signaling [17,18]. Notably, all three phosphatidylinositol 3-kinase-related kinases (PIKKs), ATM, ATR, and DNA-PK, can act as apical kinases in hyperthermia-induced DDR [19]. Recently, PARP1 was suggested to be the part of hyperthermia-induced DDR, as evident from the increased level of PAR in hyperthermia-treated HeLa and CHO-K1 cells [20]. To test whether sPARroW can be applied to monitor hyperthermia-induced DDR, we conducted a hyperthermia (42 °C, 30 min) experiment with cells transiently expressing sPARroW, also adding caffeine pretreatment for PIKK inhibition [21]. We used the ratiometric imaging approach described above, and monitored both the accumulation of the fluorescence signal in the nuclei and increased FRET efficiency after the treatment (Figure 4A,B). The observed effects were reversible and decreased rapidly to the level of control untreated cells (data not shown). These findings were complemented by traditional immunostaining for PAR (Figure 4C,D).

## 3. Discussion

FRET between two fluorescent proteins is perhaps the most often used principle of genetically encoded sensors due to a relatively simple design. Compared to turn-on sensors based on circularly permuted fluorescent proteins, FRET sensors possess much lowed dynamic range but better quantifiable ratiometric and FLIM response. FLIM is independent of fluorophore concentration, photobleaching and excitation intensity, and thus is more robust than intensity-based techniques. Cyan and yellow fluorescent proteins represent the most popular FRET pair. Here we used advanced variants of these proteins—Turquoise2 and Venus. As the proposed design of sPARroW is simple and modular, we expect that other FRET-pairs of fluorescent proteins can be used similarly.

We chose the WWE domain as a targeting part for several reasons. First, it recognizes iso-ADPR, which is characteristic only for PAR ensuring high specificity of the sensor response. Second, iso-ADPR is present in all types of PARylation—oligo(ADP-ribosylation), poly(ADP-ribosylation), and branched poly(ADP-ribosylation)—at quantities that are roughly proportional to the average size of PAR molecule. Thus, we do not expect any strong bias of the sensor sensitivity toward specific PAR modifications. Finally, affinity of WWE domain binding is intermediate among all of the PAR-binding domains (K_d_ = 372 nM [12]), which makes it less likely to interfere with physiological processes by excluding endogenous PAR-binding proteins from PAR-protein complexes.

In our experiments, we used a variety of PAR-induced stimuli—H_2_O_2_, UV laser illumination, or combination of hyperthermia and caffeine. In all cases sPARroW successfully detected PAR at the level of single live cells, enabling direct visualization of dynamics and cell-to-cell heterogeneity of PAR biosynthesis.

## 4. Materials and Methods

### 4.1. Genetic Constructs

The WWE domain sequence was amplified using human cDNA as a template. For expression in mammalian cell culture, Turquoise2-WWE, or Venus-WWE ORFs (Open Reading Frames) were cloned into C1 mammalian expression vector scaffold with the pCMV-IE promoter (Clontech) using restriction enzyme-based or self-assembly cloning [22]. Full protein sequence of the constructs is provided in Appendix A. Point mutagenesis to introduce R163A mutation in the WWE domain (numbering according to the full RNF146 protein sequence) was performed using self-assembly cloning. NLSx3 sequence was added to the 3′-end of the ORF by a sequence of step-out PCR reactions. Synthetic DNA oligonucleotides for cloning and mutagenesis were purchased from Evrogen (Evrogen, Moscow, Russia). Polymerase chain reactions (PCRs) were carried out using the PTC-100 thermal cycler (MJ Research, Waltham, MA). Purification of PCR products and products of digestion was performed by gel electrophoresis and extraction using the Cleanup Standard Kit (Evrogen, Moscow, Russia). For bacterial expression, Turquoise2-WWE and Venus-WWE ORFs were cloned into the pQE-30 vector (Qiagen, Hilden, Germany) using *BamH1* and *HindIII* restriction sites or by self-assembly cloning. Restriction endonucleases were purchased from Sibenzyme (Novosibirsk, Russia). For stable line development, Turquoise2-WWE, Venus-WWE ORFs were cloned into the pRRL.SIN.EF1.WPRE vector (a gift from Didier Trono, http://tronolab.epfl.ch) with a modified poly-linker sequence using *Age1* and *Sal1*.

### 4.2. Cell Culture and Treatments

For transient expression experiments, human osteosarcoma U2OS cell line was used. Cells were plated at 5 × 10^4^ cells per 35 mm glass-bottomed culture dish (Fluorodish, WPI, Sarasota, FL, USA) and grown in the DMEM medium (PanEco, Moscow, Russia) with 10% (*v/v*) FBS (fetal bovine serum; Sigma, Merck, Darmstadt, Germany) containing 50 U/mL penicillin and 50 μg/mL streptomycin (PanEco), 2 mM L-glutamine (PanEco) at 37 °C and 5% CO_2_ for 24 h before transfection. Two components of the sensor were introduced into the cells by co-transfection with two plasmid vectors. Transient transfections were performed with the FuGene 6 reagent (Promega, Madison, WI, USA), according to the manufacturer’s protocol using 2 μg of plasmid DNA (1 μg of each component of bipartite sensor) per transfection.

For PARP inhibitor treatment, cells were incubated with 10 µM PJ34 for 2.5 or 4 h prior to H_2_O_2_ treatment in HEPES-buffered DMEM (PanEco) supplemented with 10% (*v/v*) FBS at 37 °C in 5% CO_2_ atmosphere.

For heat shock-induced PAR analysis, U2OS cells with transient expression of sPARroW were treated with 8 mM caffeine for 3 h and then were immersed in precision controlled water-bath at 42 °C for 30 min. Immediately after incubation cells were subjected to live cell imaging using a DMIRE2 TCS SP2 confocal microscope.

Stable cell line development. Twenty four hours prior to transfection 1.5 × 10^6^ HEK-293T cells were seeded into 60 mm cell culture dishes in 3 mL of DMEM with 10% of FBS. One hour before transfection, the medium was changed to 1.3 mL of Opti-MEM. Of Opti-MEM 350 μL was mixed with 20 μL of in-house transfection reagent (Transfectin) and incubated for 5 min. Two micrograms of the pRRL plasmid, carrying either the Venus-WWE or Turquoise-WWE sequence, were added in 350 μL of Opti-MEM together with packaging plasmids pR8.91, 2.0 μg and pMD.G 0.6 μg. Plasmid solution was slowly added to Transfectin-Opti-MEM mixture, and carefully mixed. After 15 min of incubation the mix was added to the cells dropwise.

After 6 h of incubation, the medium was changed back to DMEM with 10% of FBS. Virus-containing medium was collected after 48 h, filtered through 0.45-µm filter, and immediately used to infect U2OS cells.

To produce stable cell lines expressing both Venus-WWE and Turquoise2-WWE, co-infection was performed. U2OS cells were plated at 5 × 10^4^ cells per 35 mm cell culture dish 24 h prior to the infection. For the infection, media were replaced to 1 mL of each viral media with addition of 1 mL of fresh DMEM with 10% of FBS. For the single transduction only 1 mL of chosen viral media was used. After 6 days of cultivation, cells expressing both mTurquoise and mVenus were sorted using FACS (BD FACSAria III cell sorter (BD Biosciences, San Jose, CA, USA) equipped with 100 uM nozzle, FITC channel for mVenus, AmCyan channel for mTurquoise)

### 4.3. Live Cell Imaging

Live cell imaging was performed 24–48 h after transfection in Hepes-buffered DMEM (PanEco) supplemented with 10% (*v/v*) FBS at 37 °C. Cells were imaged using a Leica confocal inverted microscope DMIRE2 TCS SP2 (Leica, Wetzlar, Germany); HCX PL APO lbd.BL 63.0 × 1.40 OIL objective, 458 nm excitation/470–490 nm emission for donor, 514 nm excitation/525–545 nm emission for acceptor, and 458 nm excitation/525–545 nm emission for FRET.

### 4.4. Ratiometric FRET Analysis

Fluorescent signal intensity was measured using ImageJ software for each channel. FRET efficiency was calculated according to the following equation [23] (as used in the Leica confocal software LCS ver. 2.61, “FRET sensitized emission” application):FRET_eff = (B − b × A − (c − a × b) × C)/C*A*—donor emission (by donor excitation) = donor channel;*B*—acceptor emission (by donor excitation) = FRET channel;*C*—acceptor emission (by acceptor excitation) = acceptor channel.Parameters *A*, *B,* and *C* were measured in cells containing both fluorescent proteins.Correction coefficients *a*, *b,* and *c* were calculated by cells containing only one fluorescent protein:*a*—(by acceptor only measurement) donor emission (by donor excitation) divided by acceptor emission (by acceptor excitation);*b*—(by donor only measurement) acceptor emission (by donor excitation) divided by donor emission (by donor excitation);*с*—(by acceptor only measurement) acceptor emission (by donor excitation) divided by acceptor emission (by acceptor excitation).

### 4.5. Fluorescence Lifetime Imaging Microscopy (FLIM)

Two-photon fluorescence microscopy and FLIM measurements were carried out with an LSM 880 (Carl Zeiss, Oberkochen, Germany) inverted laser scanning microscope equipped with a time-correlated single photon counting (TCSPC) system Simple-Tau 152 (Becker and Hickl GmbH, Berlin, Germany) Two-photon excitation of Turquiose2 was performed with a Mai Tai (Spectra Physics, Santa Clara, CA, USA) Ti:Sa femtosecond laser with 80 MHz repetition rate and a pulse duration of 100 fs at 860 nm; fluorescence was detected using a 450–490 nm emission filter. An oil immersion C Plan-Apochromat 40×/1.2 objective was used for all image acquisitions with an image size of 213 μm × 213 μm (512 pixels × 512 pixels). The cells were maintained at 37 °C and 5% CO_2_.

The first image was made before and the next immediately after 30 s irradiation with 0.2% of 405-nm laser power within the irradiated area, and then 9 min after. Fluorescence lifetime of Turqouise2 (FRET donor) was measured for every image pixel and displayed by means of intensity-weighted pseudocolors (i.e., pixels with higher intensity appear brighter).

### 4.6. Protein Purification

Fluorescent proteins were expressed in *Escherichia coli* XL1 Blue strain (Invitrogen) in the Luria Bertani (LB) medium. Cell cultures were pelleted by centrifugation, re-suspended in PBS pH 7.4 (10 mM phosphate buffer, 137 mM NaCl, and 2.7 mM KCl) and sonicated Recombinant protein purification was carried out using Talon metal-affinity resin (Clontech, Takara Bio USA, Mousntain View, CA, USA). Absorption and excitation–emission spectra were measured using Cary 100 UV/VIS Spectrophotometer and Varian Cary Eclipse Fluorescence spectrophotometer (Varian Inc., Palo Alto, CA, USA). Synthetic PAR was from Trevigen (#4336-100-01, Trevigen, Gaithersburg, MD, USA).

### 4.7. Indirect Immunofluorescence

Cells were grown on microscope slides, treated as described, fixed, and permeabilized in cytoskeletal (CSK) buffer (10 mM PIPES, pH 7.0, 100 mM NaCl, 1.5 mM MgCl_2_, and 300 mM sucrose) with 1% PFA and 2.5% Triton X-100 for 15 min at room temperature. Fixed cells were washed three times for 5 min in PBS. After washing, the cells were blocked with 1% BSA in PBS for 30 min to reduce non-specific antibody binding and incubated with a primary antibody against PAR (Trevigen, # 4336-BPC-100) in PBS with 1% BSA for 1 h at room temperature. Then after three washes for 5 min with PBS, the samples were incubated with Alexa Fluor 488-conjugated secondary antibodies (Molecular Probes/Life Technologies, Carlsbad CA, USA). Cell nuclei were stained with DAPI in PBS for 5 min at room temperature. After washing in PBS and distilled water, the samples were mounted using Dako fluorescent mounting medium (Life Technologies, Carlsbad CA, USA) and analyzed using a Zeiss AxioScope A.1 fluorescence microscope (Manu. Image processing and measurement of total fluorescence intensity of immunostained PAR was carried out with CellProfiler software (open source software, https://cellprofiler.org/).

## 5. Conclusions

In summary, we describe a novel genetically encoded fluorescence-based molecular tool for monitoring of PAR dynamics in live cells. Its design employs PAR-binding property of a recently characterized WWE domain of human RNF146 protein. We achieved live-cell imaging of both accumulation and depletion of PAR after H_2_O_2_ treatment and localized laser irradiation. The mechanism of PAR sensing of our molecular indicator does not require any exogenous labels or cofactors, which makes it potentially applicable on the intact organism level in conjunction with two-photon microscopy. Furthermore, the versatility of possible readouts (translocation vs. ratiometric FRET imaging vs. FLIM-FRET imaging) makes it readily applicable in various microscopy setups, from simple epi-fluorescence microscopy to sophisticated time-resolved imaging.

## Figures and Tables

**Figure 1 ijms-21-05004-f001:**
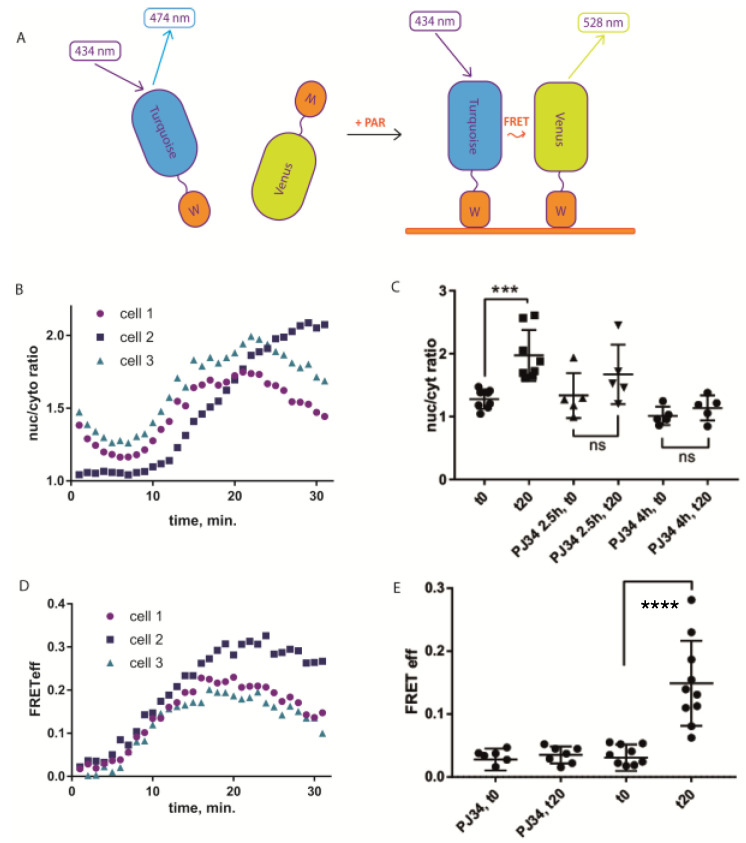
Sensor for PARrelying on WWE (sPARroW) response to PAR accumulation stimulus. (**A**) Principle of action of the sensor. (**B**) Time-lapse detection of sPARroW fluorescence distribution between nucleus and cytoplasm after H_2_O_2_ treatment. (**C**) End-point detection of sPARroW fluorescence distribution between nucleus and cytoplasm after H_2_O_2_ treatment with and without pretreatment with the PARP inhibitor (squares—w/o inhibitor, triangles—2.5 h pre-incubation with PJ34, circles—4h pre-incubation with PJ34. (**D**) Time-lapse detection of sPARroW Förster resonance energy transfer (FRET) efficiency after H_2_O_2_ treatment. (**E**) End-point detection of sPARroW FRET efficiency after H_2_O_2_ treatment with and without pretreatment with the PARP inhibitor. (B-E) U2OS cells were imaged with confocal scanning microscope in a temperature-controlled chamber set to 37 °C. Mean fluorescence intensity signals from the donor, acceptor, and FRET channels were used to calculate FRET efficiency in selected regions of interest according to the formula in the Methods section. Mean values ± SD are depicted. Statistical significance was assessed with a one-way-ANOVA and Tukey post-test. The level of significance is given with ns = not significant, *** *p* < 0.001, and **** *p* < 0.0001.

**Figure 2 ijms-21-05004-f002:**
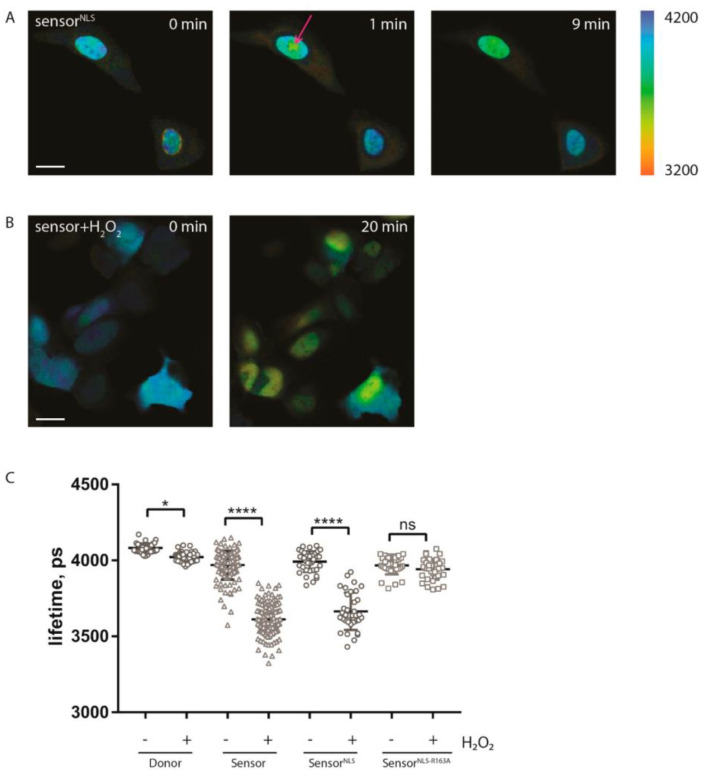
FLIM-based monitoring of sPARroW response. (**A**) U2OS cells, transiently expressing sPARroW^NLS^ and locally irradiated with a 405 nm laser; irradiation site is indicated with an arrow. Images are pseudocolored according to fluorescence lifetimes. Scale bar 25 µm. (**B**) U2OS cells, transiently expressing sPARroW, before and after treatment with 100 μM H_2_O_2_. Scale bar 25 µm. (**C**) U2OS cells transiently expressing WWE-donor, sPARroW, sPARroW^NLS^ or sPARroW^NLS-R163A^ and treated with H_2_O_2_ for 20 min. Quantitative analysis of fluorescence lifetimes of at least 34 cells. Mean values ± SD are depicted. Statistical significance was assessed with a one-way-ANOVA and Tukey posttest. The level of significance is given with ns = not significant, * *p* = 0.014, and **** *p* < 0.0001.

**Figure 3 ijms-21-05004-f003:**
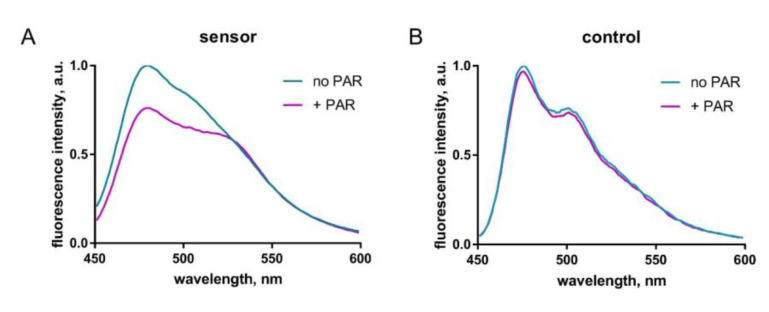
FRET increase after addition of synthetic PAR to the purified sPARroW in solution. (**A**) sPARroW emission spectra before and after addition of PAR. (**B**) Control emission spectra of mixed donor and acceptor without the WWE domain (negative control) before and after addition of PAR.

**Figure 4 ijms-21-05004-f004:**
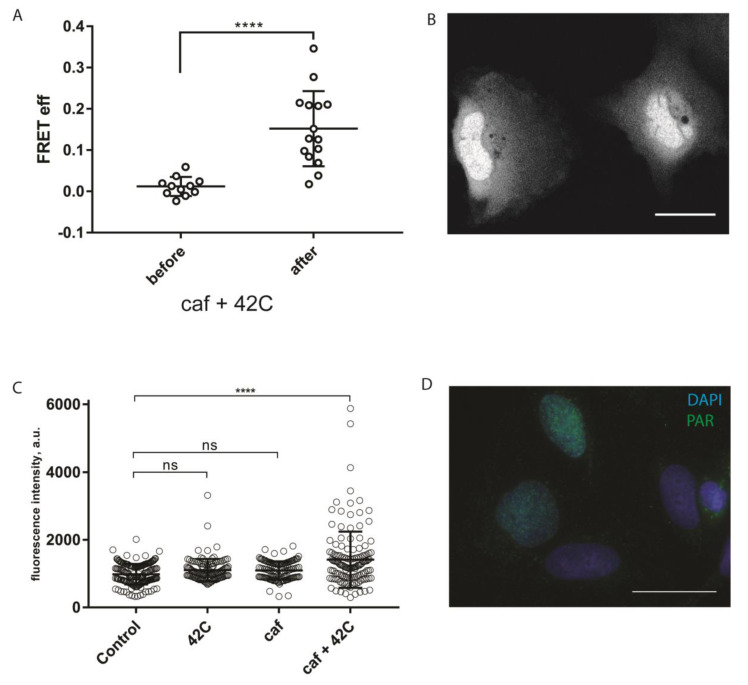
PAR accumulation after caffeine and heat shock treatment can be monitored with sPARroW. (**A**) FRET efficiency increases in U2OS cells expressing sPARroW after caffeine treatment followed by 30 min heat shock at 42 °C. At least 10 cells were analyzed per experiment. Scale bar 25 µm. (**B**) Nuclear accumulation of the fluorescence signal from sPARroW after caffeine/heat shock treatment. (**C**) Immunofluorescence analysis for PAR accumulation, U2OS cells treated with caffeine, heat shock, and both stimuli. (**D**) Confocal microphotograph of caffeine/heat shock treated cells stained with DAPI (blue) and anti-PAR antibodies (green). Scale bar 25 µm. **(A**,**C**)—Mean values ± SD are depicted. Statistical significance was assessed with a one-way-ANOVA and Tukey posttest. The level of significance is given with ns = not significant, **** *p* < 0.0001.

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
