# Peer review of "Genetically Encoded Fluorescent Sensor for Poly-ADP-Ribose"

_ijms, 2020, doi:10.3390/ijms21145004_

Round 1

Reviewer 1 Report

Poly-(ADP-ribosyl)-ation is a reversible post-translational modification and important regulator for a variety of cellular processes. Despite the importance of PARylation and the wide interest therein tools for real time-monitoring intracellular poly-ADP-ribose (PAR) levels in living cells are scarce. Within the present study, Serebrovskaya et al now present a genetically encoded FRET-based sensor for PAR localisation (sPARroW), accumulation and degradation in live cells. This sensor can certainly add to the tool box for investigating PARylation dynamics in live cell assays. The study is well designed and conceived, however, some more controls are required to fully justify the findings. Possible limitations of the sensor should be addressed in more detail.

Major points:

  • The authors use different approaches to induce PARylation in live cells (i.e. H2O2, hyperthermia, laser irradiation) and monitor the response of sPARroW in live cells. However, it is necessary to prove that the experimental treatments induced PARylation by an alternative, already established assay not just the novel sensor alone.
  • How does differential stochiometric expression of the FRET pair constructs affect read-out (FRET efficiency). Assuming that both WWE fusion proteins have the same affinity to PAR, excess of one fusion protein could reduce FRET efficiency despite increased PARylation. This could be an issue when directly (and quantitatively) comparing PARylation levels within different cells.
  • Have the authors determined what is the minimum PAR length that can be detected, are their steric hinderances for the fusion protein to bind short PAR sequences. In this case the sensor will preferentially detect specific PAR modifications, possibly missing / underestimating others.
  • When using laser irradiation, did the authors assess whether strong irradiation with the 405 nm laser does not affect the Turquoise2 molecule/fluorescence (this likely adsorbs some of the laser irradiation)

Minor points:

  • Please specify all abbreviations in the abstract (e.g. PAR, WWE)
  • Some of the graphs are not clearly labelled. E.g. labelling of the x-axis in 1C, 1E suggests that either PJ34 or sPARroW was used not PJ34 + sPARroW. Similarly, for fig 4A it is not clear whether “after” refers to after caffeine or after heat shock.

Author Response

Response to Reviewer 2 Comments

Poly-(ADP-ribosyl)-ation is a reversible post-translational modification and important regulator for a variety of cellular processes. Despite the importance of PARylation and the wide interest therein tools for real time-monitoring intracellular poly-ADP-ribose (PAR) levels in living cells are scarce. Within the present study, Serebrovskaya et al now present a genetically encoded FRET-based sensor for PAR localisation (sPARroW), accumulation and degradation in live cells. This sensor can certainly add to the tool box for investigating PARylation dynamics in live cell assays. The study is well designed and conceived, however, some more controls are required to fully justify the findings. Possible limitations of the sensor should be addressed in more detail.

Major points:

  • The authors use different approaches to induce PARylation in live cells (i.e. H2O2, hyperthermia, laser irradiation) and monitor the response of sPARroW in live cells. However, it is necessary to prove that the experimental treatments induced PARylation by an alternative, already established assay not just the novel sensor alone.

We used standard immunostaining to verify sensor response. These data are already present in Fig. 4 and related text (lines 152-153) “These findings were complemented by traditional immunostaining for PAR (Fig. 4C, D)”. In the revised version, we added the following text (lines 100-104) and Supplementary Figure S5:

“To verify sPARroW response by an independent method, we used standard immunostaining with commercial polyclonal antibodies against PAR. Upon treatment with 100 µM H2O2, we detected PAR accumulation in individual cell nuclei that was abolished by pretreatment with PJ34 inhibitor (Fig. S5). This behavior corresponded well to the sPARroW-based results.”

  • How does differential stochiometric expression of the FRET pair constructs affect read-out (FRET efficiency). Assuming that both WWE fusion proteins have the same affinity to PAR, excess of one fusion protein could reduce FRET efficiency despite increased PARylation. This could be an issue when directly (and quantitatively) comparing PARylation levels within different cells.

As both parts of the sensor have the same WWE recognition domain, and also Venus and Turquoise2 are very similar to each other (differing by a few mutations), we assumed that they have the same affinity to PAR. In transient transfections we used 1:1 ratio of the donor and acceptor. Indeed, because of stochastic variations this ratio can be somewhat different in individual cells - an issue that we addressed by measuring FRET signal in many cells. At the same time, we found no correlation between FRET efficiency and donor/acceptor ratio under

conditions used. We now discuss this in the text (lines 97-100) and new Supplementary Figure S4:

“Potential advantage of using stable expression is that cells with relatively low variability of donor and acceptor expression levels can be selected by FACS-sorting. However, we found that even with transient expression, donor/acceptor ratio is mostly uniform between individual cells and does not correlate with FRET efficiency (Fig. S4).”

  • Have the authors determined what is the minimum PAR length that can be detected, are their steric hindrances for the fusion protein to bind short PAR sequences. In this case the sensor will preferentially detect specific PAR modifications, possibly missing / underestimating others.

According to literature data, WWE domain recognizes iso-ADP-ribose (iso-ADPR), the smallest internal PAR structural unit. iso-ADPR is present in all types of PARylation at quantities proportional to the average size of PAR molecule. Thus, we do not expect any strong bias of the sensor toward specific PAR modifications. Obviously, there should be at least two close iso-ADPR units in the particular PAR molecule for efficient FRET between bound WWE-Turquoise2 and WWE-Venus.

In our experiments in vitro, we detected FRET between WWE-Turquoise2 and WWE-Venus interacting with commercially available synthetic PAR from Trevigen. According to supplier’s description “the PAR chain length ranges from 2 to 300 monomers”. However, we did not measure the minimum PAR length that can be detected by sPARoW experimentally.  

To address the concern raised we added to the manuscript the following discussion (lines 171-176):

“We chose WWE domain as a targeting part for several reasons. First, it recognizes iso-ADPR, which is characteristic only for PAR ensuring high specificity of sensor response. Second, iso-ADPR is present in all types of PARylation – oligo(ADP-ribosylation), poly(ADP-ribosylation), and branched poly(ADP-ribosylation) – at quantities that are roughly proportional to the average size of PAR molecule. Thus, we do not expect any strong bias of the sensor sensitivity toward specific PAR modifications.”

  • When using laser irradiation, did the authors assess whether strong irradiation with the 405 nm laser does not affect the Turquoise2 molecule/fluorescence (this likely adsorbs some of the laser irradiation)

Laser irradiation was localized and not too strong (0.02% of power of 405 nm line), it resulted in no significant bleaching of Turquoise2 signal. In any case, photobleaching of a dye usually does not lead to the change of its fluorescence lifetime.  

Minor points:

  • Please specify all abbreviations in the abstract (e.g. PAR, WWE)

Line 22: “fluorescent sensor for poly-(ADP-ribose) (PAR) based on Förster Resonance Energy Transfer”. As we can see from the literature, WWE is usually not treated as an abbreviation.

  • Some of the graphs are not clearly labelled. E.g. labelling of the x-axis in 1C, 1E suggests that either PJ34 or sPARroW was used not PJ34 + sPARroW. Similarly, for fig 4A it is not clear whether “after” refers to after caffeine or after heat shock.

We modified the figures according to reviewer’s suggestion.

Reviewer 2 Report

It is not clear in which experiments the authors used stable transfection. The overall Results and Discussion section do not even mention the use of stable transfection, but at the same time, describes stable transfection in detail in the Methods section. Furthermore, the methods do not specify where pRRL.SIN.EF1.WPRE comes from.

Figure 1 does not describe the exact spectral conditions under which the data presented were recorded. It is not known whether the nucl/cyto ratio in Figure 1B is given for a donor, acceptor, or FRET signal.

It is not known to what extent the donor or acceptor is expressed in the cells. This is critical for FRET measurement, as the FRET signal depends on the donor/acceptor ratio. Therefore, the expression of the bipartite sensor from two separate vectors hides a theoretical error in FRET measurement. There is a vector construct that provides an equal expression of two proteins. It would have been luckier to use such vector design. How FRET depends on the donor/acceptor ratio, which is independent of the amount of PAR, is not given or measured.

I did not receive and cannot access the supplementary material; however, the supplementary material referred to in the MS contains several figures.

It is not clear why it was necessary to use an NLS signal for the sensors if the sensors were concentrated in the nucleus even without NLS. What was the reason for the concentration in the nucleus without NLS? PAR does not form in the cytoplasm.

It is written that the fluorescence lifetime of only the donor sample is also reduced. Is it not considered possible that homo-FRET underlies the phenomenon? It would be good to show the change in the lifetime of the acceptor-only sample in Figure 2 as well.

It would be good to provide a literature reference for FRET calculation and not just reference the software used.

Author Response

Response to Reviewer 1 Comments

  • It is not clear in which experiments the authors used stable transfection. The overall Results and Discussion section do not even mention the use of stable transfection, but at the same time, describes stable transfection in detail in the Methods section. Furthermore, the methods do not specify where pRRL.SIN.EF1.WPRE comes from.

The stable line experiment is mentioned in the text as follows (lines 95-98):

“It was also possible to detect H2O2-dependent FRET efficiency change in U2OS cell line stably expressing sPARroW (Fig. S3). Potential advantage of using stable expression is that cells with relatively low variability of donor and acceptor expression levels can be selected by FACS-sorting.”

Information about lentivirus vector was added (lines 199-201):

“For stable line development, Turquoise2-WWE, Venus-WWE ORFs were cloned into pRRL.SIN.EF1.WPRE vector (a gift from Didier Trono, http://tronolab.epfl.ch) vector with modified poly-linker sequence using Age1 and Sal1.”

  • Figure 1 does not describe the exact spectral conditions under which the data presented were recorded. It is not known whether the nucl/cyto ratio in Figure 1B is given for a donor, acceptor, or FRET signal.

Nuc/cyto ratio is given for acceptor signal intensity. Lines 84-87:

“To test its response to PAR-inducing stimuli, we first analyzed subcellular distribution of sPARroW and found that it accumulated in the nuclei of H2O2-treated cells, reaching peak nuc/cyto ratio (as calculated by acceptor signal intensity) 25 minutes after addition of 100 µM H2O2.”

  • It is not known to what extent the donor or acceptor is expressed in the cells. This is critical for FRET measurement, as the FRET signal depends on the donor/acceptor ratio. Therefore, the expression of the bipartite sensor from two separate vectors hides a theoretical error in FRET measurement. There is a vector construct that provides an equal expression of two proteins. It would have been luckier to use such vector design. How FRET depends on the donor/acceptor ratio, which is independent of the amount of PAR, is not given or measured.

From our experience with dual-promoter or IRES-containing vectors, the level of expression of two proteins is also not equal in these systems. The potential advantage of using one of these systems would be that they could give more consistent ratio between donor and acceptor expression levels. However, we decided not to use these systems thinking of testing different combinations of donor/acceptor constructs. The issue of uneven donor/acceptor ratio is easily addressed by measuring FRET signal in many individual cells. We also found that transfection with two plasmids at 1:1 gives rather narrow range of donor/acceptor signals ratio. We added the following phrase (lines 98-100) and Supplementary Figure 4:

“However, we found that even with transient expression, donor/acceptor ratio is mostly uniform between individual cells and does not correlate with FRET efficiency (Fig. S4).”

For stable cell line experiments, we FACS-sorted the cells that fell within a narrow range of Venus and Turquoise2 signal levels.

  • I did not receive and cannot access the supplementary material; however, the supplementary material referred to in the MS contains several figures.

We are very sorry for this technical mistake! Now Supplementary Figures are submitted.

  • It is not clear why it was necessary to use an NLS signal for the sensors if the sensors were concentrated in the nucleus even without NLS. What was the reason for the concentration in the nucleus without NLS? PAR does not form in the cytoplasm.

In the absence of NLS or any other mechanism for preferential nuclear localization or exclusion, fluorescent protein-based constructs below certain size cutoff are mostly evenly distributed across the cell, and thus can access the nucleus too – presumably by passive diffusion through the nuclear pore [Wang R, Brattain MG. The maximal size of protein to diffuse through the nuclear pore is larger than 60 kDa. FEBS Lett. 2007; 581:3164-3170]. PAR is formed predominantly in the nucleus of the cells, so we assume that PAR binding is the mechanism for the shift from even distribution towards preferentially nuclear localization. Nuclear accumulation is observed for other fluorescent protein-based PAR-detecting systems [Krastev DB, Pettitt SJ, Campbell J, et al. Coupling bimolecular PARylation biosensors with genetic screens to identify PARylation targets. Nat Commun. 2018; 9:2016]. The rationale behind using NLS in our constructs was to minimize the potential error that could be caused by simultaneous change of FRET together with the local concentration of the sensor.

Also, we occasionally observed higher-than-usual nuc/cyto ratio at t0 (before PARylation-inducing stimulus) in some cells. We attribute this to the basal PAR accumulation caused by the intrinsic state of the cells, and/or cell culture conditions.

  • It is written that the fluorescence lifetime of only the donor sample is also reduced. Is it not considered possible that homo-FRET underlies the phenomenon? It would be good to show the change in the lifetime of the acceptor-only sample in Figure 2 as well.

Yes, we considered possibility of homo-FRET since the close proximity of many donor molecules interacting with PAR is expected in this case. However, many papers and books point out that homo-FRET can be detected by polarization microscopy only, since homo-FRET does not affect fluorescence lifetime, intensity, and spectrum [see e.g.: Chan FT, Kaminski CF, Kaminski Schierle GS. HomoFRET fluorescence anisotropy imaging as a tool to study molecular self-assembly in live cells. Chemphyschem. 2011, 12, 500-509]. In all our experiments, we did not measure fluorescence lifetime of acceptor as this value is not changed in FRET.

  • It would be good to provide a literature reference for FRET calculation and not just reference the software used.

We added the reference:

  1. Wouters FS, Verveer PJ, Bastiaens PI. Imaging biochemistry inside cells. Trends Cell Biol. 2001;11(5):203-211. doi:10.1016/s0962-8924(01)01982-1

Round 2

Reviewer 1 Report

The authors have addressed all comments adequately.

Reviewer 2 Report

I accept your answers.